# Evaluation of *Phoma* sp. Biomass as an Endophytic Fungus for Synthesis of Extracellular Gold Nanoparticles with Antibacterial and Antifungal Properties

**DOI:** 10.3390/molecules27041181

**Published:** 2022-02-10

**Authors:** Meysam Soltani Nejad, Neda Samandari Najafabadi, Sonia Aghighi, Elena Pakina, Meisam Zargar

**Affiliations:** 1Department of Plant Protection, Faculty of Agriculture, Shahid Bahonar University of Kerman, Kerman 7616914111, Iran; 2Department of Plant Protection, Faculty of Agriculture, Ferdowsi University of Mashhad, Mashhad 9177948978, Iran; samandarinajafabadi.neda@mail.um.ac.ir; 3Research and Technology Institute of Plant Production, Shahid Bahonar University of Kerman, Kerman 7616914111, Iran; aghighis@uk.ac.ir; 4Department of Agrobiotechnology, Institute of Agriculture, RUDN University, 117198 Moscow, Russia; pakina-en@rudn.ru

**Keywords:** nanomaterial, gold nanoparticles, *Phoma* sp., *Rhizoctonia solani* AG1-IA, *Xanthomonas oryzae* pv. *oryzae*

## Abstract

The aim of our study was to examine the different concentrations of AuNPs as a new antimicrobial substance to control the pathogenic activity. The extracellular synthesis of AuNPs performed by using *Phoma* sp. as an endophytic fungus. Endophytic fungus was isolated from vascular tissue of peach trees (*Prunus persica*) from Baft, located in Kerman province, Iran. The UltraViolet-Visible Spectroscopy (UV–Vis spectroscopy) and Fourier transform infrared spectroscopy provided the absorbance peak at 526 nm, while the X-ray diffraction and transmission electron microscopy images released the formation of spherical AuNPs with sizes in the range of 10–100 nm. The findings of inhibition zone test of Au nanoparticles (AuNPs) showed a desirable antifungal and antibacterial activity against phytopathogens including *Rhizoctonia solani* AG1-IA (AG1-IA has been identified as the dominant anastomosis group) and *Xanthomonas oryzae* pv. *oryzae*. The highest inhibition level against sclerotia formation was 93% for AuNPs at a concentration of 80 μg/mL. Application of endophytic fungus biomass for synthesis of AuNPs is relatively inexpensive, single step and environmentally friendly. In vitro study of the antifungal activity of AuNPs at concentrations of 10, 20, 40 and 80 μg/mL was conducted against rice fungal pathogen *R. solani* to reduce sclerotia formation. The experimental data revealed that the Inhibition rate (RH) for sclerotia formation was (15, 33, 74 and 93%), respectively, for their corresponding AuNPs concentrations (10, 20, 40 and 80 μg/mL). Our findings obviously indicated that the RH strongly depend on AuNPs rates, and enhance upon an increase in AuNPs rates. The application of endophytic fungi biomass for green synthesis is our future goal.

## 1. Introduction

Metal-based nanomaterials, including copper nanoparticles (CuNPs), iron (FeNPs), palladium (PtNPs), gold (AuNPs), aluminum (AlNPs), zinc (ZnNPs) and silver (AgNPs), are typically synthesized naturally or synthetically [1,2]. Gold nanoparticles are one of the most significant ones as it has several therapeutic applications [3,4] such as drug delivery system for several diseases such as cancer diagnosis and therapy [5], radiosensitizers [6], biosensors [7], bio-sensing [8], bio-imaging and environmental applications of dye degradation [9], bioremediation of toxic chemicals present in the environment [10] and muscle tissue engineering [11].

Currently, for the biosynthesis of AuNPs, the extracellular method is the most common method [12,13]. Specifically, the typical synthesis methods for AuNPs are physical (UV irradiation, laser ablation and plasma synthesis), chemical (citrate synthesis, wet chemical synthesis and chemical reduction) and physicochemical (sono-chemical and sono-electrochemical) methods [14]. Endophytes fungi are potential sources of bioactive metabolites that have huge usages in the medical fields, health care, farming systems, industry, biology and its derivatives reveal anticancer, immunomodulatory, antitubercular, antiviral and antidiabetic activities [15,16]. Hence, evaluating endophytic fungi which inhabit various plant varieties by the therapeutic characterizations would provide opportunities to find out new metabolites by the specific bioactivity [17]. These fungi interact with the host plant, and in turn, the plants to some extent modulate the metabolic process of these endophytes to produce molecules that could manifest protective functions towards the microbe and the host [18].

Using entophytic fungi for biosynthesis of AuNPs is beneficial as they are cultured easier compared to other microorganisms, and they can also secrete major amounts of protein [19]. Manjunath et al. [20] applied *Penicillium citrinum* isolated from brown algae for the synthesis of gold nanoparticles for its antioxidant activity. Fatima et al. used a novel phosphate solubilizing fungus *Bipolaris tetramera* isolated from rhizospheric region for extracellular synthesized AuNPs, while Bhambure et al. [21] reported the ability of *Aspergillus niger* for the extracellular synthesis of these particles. Lee et al. [22], biosynthesized spherical, triangle, hexagonal and rod-shaped gold nanoparticles by the using extracts of *Inonotus obliquus* (*Chaga mushroom*). Xie et al. [23], used mycelia-free spent medium of filamentous fungus (*Aspergillus niger)* for the biosynthesis of gold nanocrystals. Previous results of the researchers indicated that the proteins on the cell wall of the fungus and the proteins in the fungal extract are the primary biological molecules [23]. Farming endeavors are defective without the use of fertilizers for crop nutrition, as well as pesticide applications for the goal of plant protection that continuously disrupt the activity, causing high economic losses [24]. The use of these types of substances has resulted in huge ecological and microbial resistance for pesticides. AuNPs-based nano-fertilizers have been developed to synchronize nutrient release with plant uptake [25]. NPs with the bacteriostatic and fungistatic activities act as the environmentally-friendly inhibitors against some plant diseases in comparison with synthetic chemical pesticides [26]; management of plant pathogenic microbes, can also be obtained with application of a NPs directly on grains, seed or foliage application to inhibit the infestation of the target plant tissue [27]. AuNPs with the antifungal and antibacterial ability has obtained less attention in comparison to medical sciences with only a few researches performed against fungal and bacterial diseases of plants [28]. Rice is the most important food in the vast part of the world, and is, also, a significant primary production in some farming systems. Rice pathogens caused by *Rhizoctonia solani* Kühn AG1 and *Xanthomonas oryzae* pv. *oryzae* are globally important pathogens destructive diseases in all rice-growing area in the world [29,30]. In this study, the extracellular biosynthesis of AuNPs is optimized by an endophytic fungus isolated from peach trees. The synthesized AuNPs were characterized through UV–Vis spectrophotometer, transmission electron microscope (TEM), Fourier transform infrared spectroscopy (FTIR) and X-ray diffraction spectroscopy (XRD). For controlling bacterial and fungal diseases in rice with emphasis on the cleaner production at a lower cost we examined the various concentrations of AuNPs as a new antifungal and antibacterial substance to suppress the pathogenic activity.

## 2. Results

### 2.1. Endophytic Fungi Isolation and Identification of Active Isolate

Several samples were collected from different parts of peach trees. Peach trees were selected with different branches, Figure 1a. Endophytic fungi were isolated from vascular tissue of central wood segments samples Figure 1b following a method utilized by Ghasemi-Sardareh et al. [31]. Figure 1c is colonies of endophytic fungus and Figure 1d shows pure culture of isolate MS7 (Meysam Soltani No. 7); Figure 1e or Figure 1f show pycnidia and conidia, respectively.

### 2.2. Molecular Identification

The rDNA-ITS (Internal Transcribed Spacer) gene was amplified by polymerase chain reaction (PCR) and BLAST search sequence program, comparing with all sequences in GenBank which showed the highest similarity to *Phoma* sp. with the E-value of 0.0 and max. Identity of 99%, the length of sequenced fragments of ITS gene was 541 bp via the accession No. MZ029391 in GenBank. The phylogenetic trees based on ITS sequences showed that our isolated MS7 is related to *Phoma* sp. and *Thermoascus crustaceus* (MN431405)*,*
Figure 2.

### 2.3. Extracellular Biosynthesis of AuNPs

For evaluation of extracellular synthesis of AuNPs activity, biomass of the isolate MS7 was prepared in potato dextrose broth (PDB) medium. The mycelia of fungus prepared as described and incubated at 28 °C with HAuCl_4_ solution for 48 h. The HAuCl_4_ ions were reduced to AuNPs through an overnight exposure to fungal biomass. Figure 3a shows a test tube containing biomass of isolate MS7 before the immersion in 1 mM HAuCl_4_ solution. The color of fungal biomass changed to red after 48 h immersion in 1 mM HAuCl_4_ solution, Figure 3b.

### 2.4. UV–Vis Analysis

After centrifuging at 4000 rpm for 10 min, pellets were discarded and 1 mL of the supernatants used for UV–Vis analysis at 450–700 nm wavelengths recorded after two different reaction times, 24 and 48 h. While untreated sample set as the reference control, the treated sample showed a prominent peak at 530 nm as the indication of AuNPs, Figure 4.

### 2.5. XRD Analysis

The XRD pattern showed four sharp peaks throughout the spectrum, Figure 5. These intense peaks were observed at 2θ definite diffraction of 38.185°, 44.215°, 64.473° and 77.444° which referred to (111), (200), (220), (311) and (331), respectively. The XRD analysis clearly showed that the AuNPs was formed by *Phoma* sp. mycelial biomass and due to the sharpness of the peaks, the nature of AuNPs was crystalline.

### 2.6. TEM and DLS Analysis

To determine the particles size distribution and shape of AuNPs, Transmission Electron Microscopy (TEM) was used on the obtained AuNPs, Figure 6 [32]. As indicated in the micrographs, the majority of the extracellular biosynthesized AuNPs was spherical in shape.

The size of gold nanoparticles were determined through dynamic light scattering (DLS) examination. The particle size distribution study indicated that numbers of particles were within the size range of 10–100 nm and Zaverage: 65.32, PDI: 0.2480, Figure 7. The DLS analysis in three modes (intensity, size and volume) was shown the average particle size of the biosynthesized displaying high monodispersity. The zeta potential analysis confirmed that the AuNPs biosynthesized are highly stable in water and do not aggregate in solution due to their negative polarity.

### 2.7. FTIR Spectra

FTIR spectra provided insight into the interactions between molecules and synthesized nanoparticles. Figure 8 shows the FTIR results of AuNPs synthesized by biomass of *Phoma* sp., which contain some strong absorption peaks at 3374.73, 2938.60, 1647.66, 1458.34 and 1384.57 cm^−1^. The absorption peak at 1647 cm^−1^ can be assigned to the amide bond of proteins from carbonyl stretching in native proteins and aliphatic amines band. The peak at 3374 cm^−1^ is assigned to the O–H stretching related to the O-H bond of H_2_O compounds and carbohydrates [33,34]. Moreover, the absorption peak at 1628 cm^−1^ is close to that reported for native proteins, which suggests that proteins are interacting with synthesized AuNPs. The FTIR results verify that the carboxyl, hydroxyl and N–H groups in the *Phoma* sp. biomass are mainly involved in the reduction of Au^3+^ ions to Au^0^ NPs [35].

### 2.8. Antibacterial and Antifungal Study

The antibacterial and antifungal properties of the synthesized AuNPs were investigated by the disk-diffusion method and well-diffusion practice for antibacterial and antifungal tests, respectively, [36] against plant disease agents. Diffusion of AuNPs caused the inhibition zones around the disks which obviously indicate the antifungal and antibacterial activities of AuNPs, Figure 9.

### 2.9. MIC Analysis

The minimum inhibitory concentration of the AuNPs was 160.6 μg/mL.

### 2.10. Inhibitory Effects of AuNPs on Sclerotia of R. solani

Figure 10 shows the evolution effects of tested AuNPs concentrations on sclerotia formation. Plates treated with 80 μg/mL AuNPs revealed the minimum number of sclerotia.

The RH of 15, 33, 74 and 93% were found for the 10, 20, 40 and 80 μg/ mL of AuNPs concentrations, respectively, Figure 11.

## 3. Discussion

Recently, green synthesis of metal nanoparticles by the application of fungi has gained special investigation due to protect ecosystem. Different AuNPs application are as bio-sensor, drug-delivery, bio-imaging, catalysts, gene delivery, antimicrobial, tumor imaging and bio-assay. Currently Fungi have been applied for the biosynthesis of the metal-based nanoparticles. Therefore, fungi have more advantages than bacteria for bioprocessing, involving green synthesis of gold nanoparticles. The so-called secretome comprise all of the secreted proteins into the extracellular space [37,38]. Many fungal secretome concentrations have been used for huge for producing proteins and the extracellular proteins and enzymes play a vital role in Au reduction and AuNP capping [39]. Having high metal toxicity resistance is the benefits of using fungi for synthesis of gold nanoparticles [40]. AuNP biosynthetic potentials in different bacteria and fungi suggest that the diminish of Au^3+^ to form protein metal nanoconjugates is a common response to toxic stress, where the enzymatic machinery needed is readily available in environmental microorganisms. Several studies on biosynthesis of AuNPs by the soluble protein extract of *Fusarium oxysporum* released that Nicotinamide adenine dinucleotide hydrogen (NADH) -dependent reductases are involved in the bioreduction process [41]. The antibacterial effectiveness of AuNPs were related to the size and dispersibility of nanoparticles. The smaller AuNPs in diameter is more applied in tissue immunology, biochemistry and high-powered microscopy. In environmental research, DNA testing and drug delivery, medium-sized AuNPs are mostly used. Larger AuNPs are used in medical, electrical and X-ray optics [42]. The mechanism for antibacterial demonstrated by the biosynthesized AuNPs is subjected to the various degree of susceptibility of bacteria. When the AuNPs come in contact with the microbe, it binds with the bacterial surface through electrostatic interaction. The distinct smaller size and proton motive force of the synthesized AuNPs enable their penetration into the bacterial cell through the membrane proteins [43]. The (AuNPs) have shown uniquely advantageous antimicrobial properties. As the least active metal, gold has very stable chemical activities, is non-toxic and has well biocompatibility. Compared with gram-positive bacteria, the structure of peptidoglycan cell walls, which gram-negative bacteria have, may cause differences in the antibacterial effect of AuNPs [44].

AuNPs characterizations were considered using UV–Vis spectroscopy, XRD, TEM, DLS and FTIR. Then, the inhibitory properties of synthesized AuNPs in various concentrations were investigated against rice diseases. We concluded that synthesized AuNPs using endophytic fungus biomass can act as a bio-reluctant and bio-capping agent, and, also, have the significant potential to inhibit the growth of sheath blight sclerot and *Xanthomonas oryzae* pv. *oryzae* in condition of in vitro. Results indicated that the synthesized AuNPs have significant antifungal and antibacterial activities. When metal ions in solution come in contact with a bacterial cell, they are generally dispersed without a specific location around the bacterial cell. On the other hand, nanoparticles that interact with the bacterial cell wall form a concentrated source of ions that continuously release ions and increase cytotoxicity [45]. AuNPs inhibited the cellular metabolic process by changing the membrane capability and diminishing adenosine triphosphate (ATP) synthase characterizations and they rejected the ribosome subunit for tRNA binding and effectively disrupted its biological process. Thus, they were found to be less harmful to membrane cells [46,47].

AuNPs are one of the most commonly used metal NPs with unique surface plasmon properties and large surface area to volume ratio that have major applications in the biomedical field. Nevertheless, chemically synthesized NPs cause toxicity in living organisms and are in conflict with their environmentally friendly nature and cost-effectiveness. Therefore, the development of an ecological synthetic pathway for the synthesis of AuNPs using natural materials is a novel research area for its effectiveness in the synthesis of non-toxic and environmentally friendly materials [48]. Various researchers have reported that environmentally safe biosynthesis of AuNPs from fungi, bacteria and plant extract by inhibitory effects on the pathogenic fungi and many bacteria. Fungi can processed with a high degree of ease, not only in the laboratory but on a large scale, since mycelium can tolerate harsh conditions in bioreactors [49]. The AuNPs can be synthesized in three ways: extracellular, fungal autolysis or intracellular. The size and distribution of fungi vary depending on the strain and experimental conditions [50].

Biosynthesis of AuNPs using fungi awards opportunities for the development of sustainable and environmentally friendly NPs with effective antibacterial properties. The effect of NPs on crops is a growing field of interest that requires careful investigation. Over recent years, engineered nanoparticles have received considerable attention as a potential candidate for the enhancement of crop performance, resistance, and disease management technologies [51]. The results of this, particular, study are at a preliminary stage for the development of nanotech-based agents. These data can be utilized as a platform for future research with the aim of commercializing these products and hoping to step towards sustainable agriculture.

The gold nanoparticles were extracellularly biosynthesized using mycelial biomass of *Phoma* sp. as an endophytic fungus for antibacterial and antifungal purposes. As treated by 1 mM HAuCl_4_ according DLS, the particle sizes were in the range of 10–100 nm. The in vitro antibacterial and antifungal properties of AuNPs were tested against rice plant pathogenic fungi and bacteria. In vitro experiment of the antifungal activity of AuNPs at concentrations of 10, 20, 40 and 80 μg/mL was conducted against rice fungal pathogen *R. solani* to reduce sclerotia formation. The in vitro results showed that the RH for sclerotia formation were respectively (15, 33, 74 and 95%) for their corresponding AuNPs concentrations (10, 20, 40 and 80 μg/mL).

## 4. Material and Methods

### 4.1. Isolation and Identification of Endophytic Fungi

For the isolation of fungi, vascular tissues of *Prunus persica* was taken from gardens located in Baft, Kerman province (southeast of Iran). Wood segments were cut from the vascular tissue samples and surface sterilized with a 1.5% sodium hypochlorite solution for one minute and then rinsed twice with sterile distilled water. Wood chips (5 mm) were cut from the healthy tissue and plated onto potato dextrose agar (PDA; Merck, Darmstadt, Germany). Petri-dishes were incubated at 28 ± 1 °C and all colonies of the endophytic fungal isolates were transferred onto fresh PDA (Potato Dextrose Agar) media for three weeks at 28 ± 1 °C. Single-spore was obtained for each isolate prior to morphological and molecular identification processes [31]. For the molecular identification of pure culture of endophytic fungus, isolates were obtained via single-spore method. The mycelia from 20-day cultures were scraped off from the surface of petri plates using a sterile scalpel. About 100 mg of fungal mycelia was grounded to powder by mortar and pestle after freezing using liquid nitrogen. The CTAB (Cetyltrimethylammonium Bromide) method was used to DNA extract genomic from mycelia [52]. For molecular identification, primers ITS5 (GGAAGTAAAAGTCGTAACAAGG) and ITS4 (TCCTCCGCTTATTGATATGC) were used to amplify ITS1, 5.8S, ITS2-rRNA from DNA [53]. For active isolation, ~5 μL PCR product was electrophoresed on 1% agarose gel (UltraPureTM Agarose, Invitrogen) containing ethidium bromide, and visualized under UV illumination. A 100-bp ladder (Gene Ruler, TMDNA Ladder Mix, Fermentas) was utilized as a molecular weight marker. PCR products were purified and sequenced by Macrogen Co, Seol, Korea, then the sequence was read and edited with BioEdit Sequence Alignment Editor. All sequences of respective gene sets were aligned using Clustal W tool in MEGA 7.0 program (Molecular Evolutionary Genetics Analysis, Biodesign Institute, Kent, OH, USA). To reconstruct phylogenetic tree, the maximum parsimony (MP) algorithm was applied. The species *Thermoascus crustaceus* (MN431405) was used as the outgroup taxon.

### 4.2. Biosynthesis of Gold Nanoparticles

For the biosynthesis of gold nanoparticles (AuNPs), the biomass of fungal isolates was cultured on potato dextrose broth with an infusion of 250 g potato and 20 g dextrose per liter of distilled water. The inoculated flasks were incubated in a rotary shaker at 28 ± 1 °C at 120 rpm for ten days. After that, the biomass was harvested and washed with double distilled water three times. The culture was centrifuged at 4000 rpm for 15 min and fungal biomass collected and used for the subsequent steps. Fresh biomass was mixed with 2 mL of 10^−3^ M aqueous Auric Chloride (HAuCl_4_), the control was without HAuCl_4_ (only biomass + distilled water). Subsequently, the reaction mixture was vigorously shaken in a shaker incubator at ambient temperature; the AuNPs appeared with a reddish color [54]. The possible mechanism for the reduction of gold nanoparticles is: *AuCl_3_ + 2e^−^* reduction to *AuCl + 2Cl^−^* and *3AuCl* change to *Au^0^ + AuCl_3_*.

### 4.3. UV–Vis Spectroscopy Analysis

The UV–Vis spectroscopy was used to authenticate the synthesis of AuNPs in the solution. It is well known that AuNPs have bill-red color, depending on the intensity and the size of AuNPs [55,56]. Both treated and untreated biomasses centrifuged for 10 min at 1000 rpm. Pellets discarded and supernatants used to monitor the UV-Vis absorbance spectra at 500–550 nm wavelengths at different times (24 and 48 h) of the reaction. While untreated sample set as the reference control, the treated sample revealed a prominent peak at 530 nm which was the indication AuNPs [57].

### 4.4. XRD Analysis

The XRD analysis provides information on the structures or crystals phase of tested sample. The synthesized AuNPs was centrifuged (at 10,000 rpm; 25 °C) for 15 min, washed with deionized water and centrifuged again for three cycles. Subsequently, the purified AuNPs were dried at room temperature and subjected to the XRD experiment. The XRD was performed on an X-ray diffractometer (Munich, Germany) [58].

### 4.5. Transmission Electron Microscopy (TEM) Analysis and Dynamic Light Scattering Spectroscopy (DLS)

Transmission electron microscopy (TEM) examination was performed to determine the shape and size of the biosynthesized AuNPs. A drop of AuNPs solution was placed on the carbon coated copper films and air-dried naturally. TEM micrographs were taken by Carl ZIESS TEM, Jena, Germany.

DLS spectroscopy was conducted to study the size distribution of freshly synthesized AuNPs using a DLS spectrometer, Cordouan Technologies Company, Bordeaux, France, under a controlled temperature (24.67 °C), 50%, laser power, up DTC (Direct torque control) position and wavelength of 657 nm.

### 4.6. Analysis of Fourier Transform Infra-Red Spectroscopy (FTIR)

For FTIR analysis, the AuNPs powder was prepared by centrifuging the synthesized AuNPs solution at 8000 rpm for five min. The product was then dried completely by incubating at 50 °C for two days, and used for the FTIR measurements through the KBr pellet method using a Bruker FTIR spectrometer TENSOR II (Bruker, Ettlingen, Germany).

### 4.7. Antibacterial and Antifungal Activity

A virulent isolate of *Xanthomonas oryzae* and *Rhizoctonia solani* AG1-IA were prepared from the Biological Control Laboratory, Department of Plant Pathology, Shahid Bahonar University of Kerman. To evaluate antibacterial properties of AuNPs, the disk diffusion method was used for plant pathogen bacteria *Xanthomonas oryzae* pv. *oryzae*. The bacterium was cultured in the plates containing nutrient agar (Merck, Darmstadt, Germany). For the inhibition effect of synthesized AuNPs, 6 mm disks drenched in 30 μL of AuNPs solution and subsequently placed on the disinfected agars and inoculated at room temperature under the laboratory hood for 24 h [59]. The negative controls were disks without AuNPs (only mycelia biomass extract solution). After 24 h, the diameter of inhibition zone caused by AuNPs and negative control was measured.

To study the antifungal activity and inhibitory effect of synthesized AuNPs on the mycelia growth of *R. solani*, an in vitro assay was performed on the PDA media treated with 30 μL of AuNPs solution. The negative control was mycelia biomass extract solution. Bioassay tests were performed through the well-diffusion method as described by Soltanzadeh et al. [56]. The PDA plates were incubated at 28 ± 1 °C for two weeks to allow the growth of *R. solani* when the control was covered completely with fungal mycelial, the radial growth of fungal mycelium was analyzed [60]. Each set of tests was repeated three times.

### 4.8. Determination of Minimum Inhibitory Concentration

In order to determine minimum inhibitory concentration (MIC) by the micro-dilution method according to Akhlaghi et al. [57], *Xanthomonas oryzae* pv. *oryzae* was cultured in LB broth medium overnight at 28 °C. About 100 μL of each AuNPs (concentration of 1000 μg/mL) was added to 100 μL Mueller Hinton broth medium to obtain 500 μg/mL of the AuNPs in the first well of each row in 96-well micro-plates. Then AuNPs were serially 2X diluted in 96-well micro-plates containing 1% of DMSO to obtain a range of concentrations from 500 to 0.48 μg/mL in a total volume of 200 μL. Subsequently, teen microliter of bacterial culture (~10^7^ CFU/mL) was inoculated into each well. The micro-plate was incubated for a day at 28 °C. The MIC was measured as the lowest concentration of the AuNPs at which no measurable growth occurred. All the experiments were performed in triplicate.

### 4.9. Evaluation of Inhibitory Effects of AuNPs on Sclerotia Formation of Rhizoctonia solani

To evaluate the antifungal effects of AuNPs against *R. solani,* four various concentrations of AuNPs suspension (10, 20, 40 and 80 µg/mL) were added to petri dishes before pouring plates with PDA media. The agar plugs with a diameter of three millimeter containing fungal mycelia were inoculated together at the center of each petri dish containing AuNPs, followed by incubation at 28 °C for three weeks. The antifungal effect of AuNPs against *R. solani* sclerotia formation was calculated after adding four concentrations of AuNPs to the PDA media content. All tests were carried out in triplicate. The inhibition rate was following formula (1). Calculated:



(1)
$$\mathrm{Inhibition}\;\mathrm{rate}\;(\mathrm{RH})\;\%\;=\;\frac{\left(R-r\right)}{R}\;\times \;100$$



The parameter RH is the inhibition rate; *R* is the weight of the sclerotia in the control dish (mg). The parameter *r* is the weight of sclerotia when treated by AuNPs (mg).

### 4.10. Statistical Analysis

Recorded data were subjected to analysis with SAS software (SAS Institute, version 9, Cary, NC, USA).

## Figures and Tables

**Figure 1 molecules-27-01181-f001:**
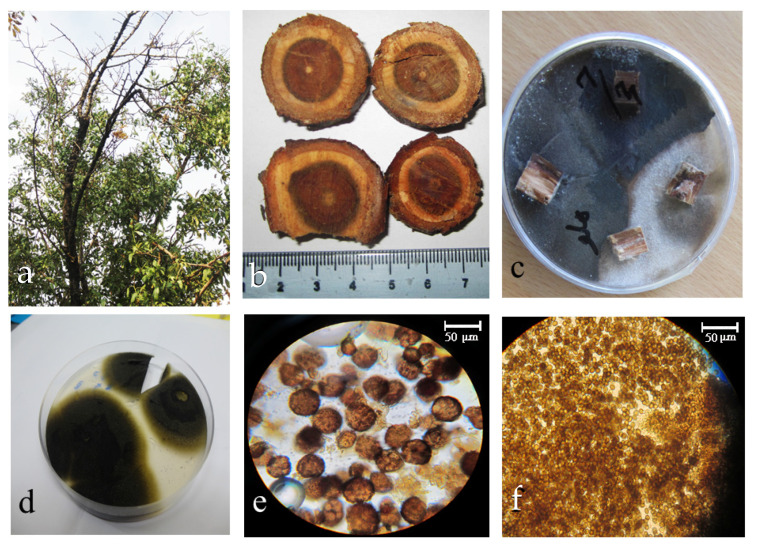
The isolation steps of endophytic fungus. (**a**) peach trees, (**b**) vascular tissues of central wood segments, (**c**) wood chips on PDA media, (**d**) the colonies with dark-brown mycelia developed of the plated, four-week-old colonies of MS7 isolate, (**e**) pycnidia of MS7 and (**f**) conidia of MS7.

**Figure 2 molecules-27-01181-f002:**
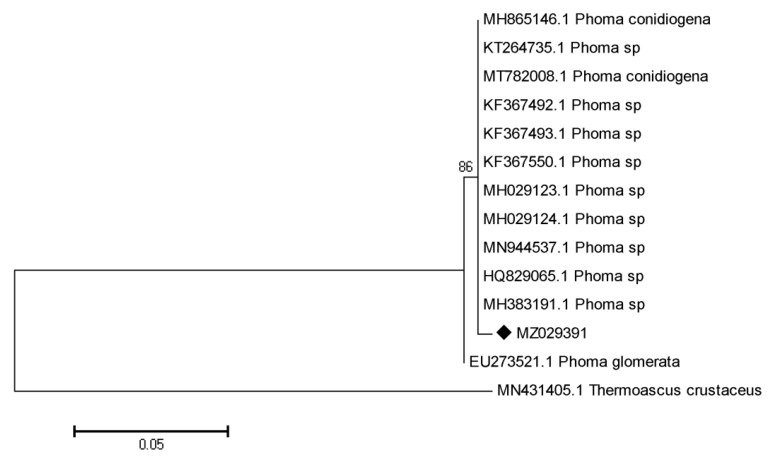
Neighbor joining analysis of MS7 and close relatives retrieved from Genbank. Most parsimonious tree based on ITS gene and phylogeny tree of MS7 isolates. Sequences BLAST search MZ029391 for the MS7 showed 99% homology with *Phoma* sp. and *Thermoascus crustaceus* was used as outgroup.

**Figure 3 molecules-27-01181-f003:**
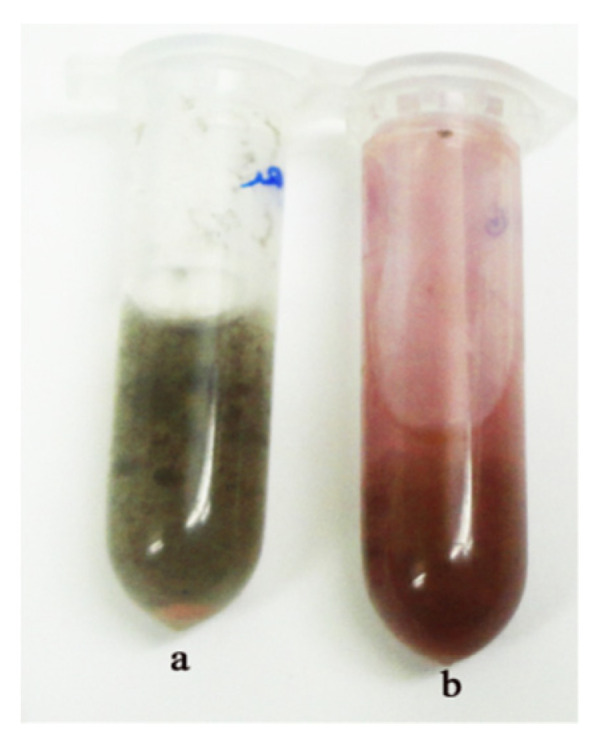
Synthesized AuNPs in a colloidal dispersion by *Phoma* sp. Mycelia biomass before (**a**) and after (**b**) exposure to HAuCl_4_ after 48 h.

**Figure 4 molecules-27-01181-f004:**
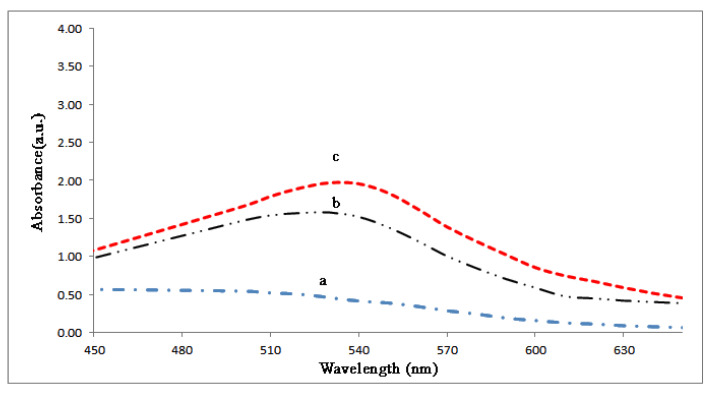
UV–Vis spectra of extracellular synthesized AuNPs by *Phoma* sp. mycelial biomass at different times: a, b and c are control, 24 h and 48 h after biosynthesis reaction, respectively.

**Figure 5 molecules-27-01181-f005:**
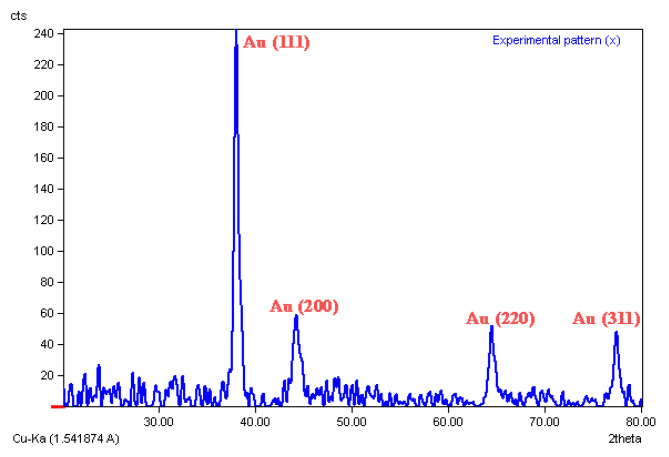
X-ray diffraction (XRD) spectrum of AuNPs synthesized by the biomass of *Phoma* sp.

**Figure 6 molecules-27-01181-f006:**
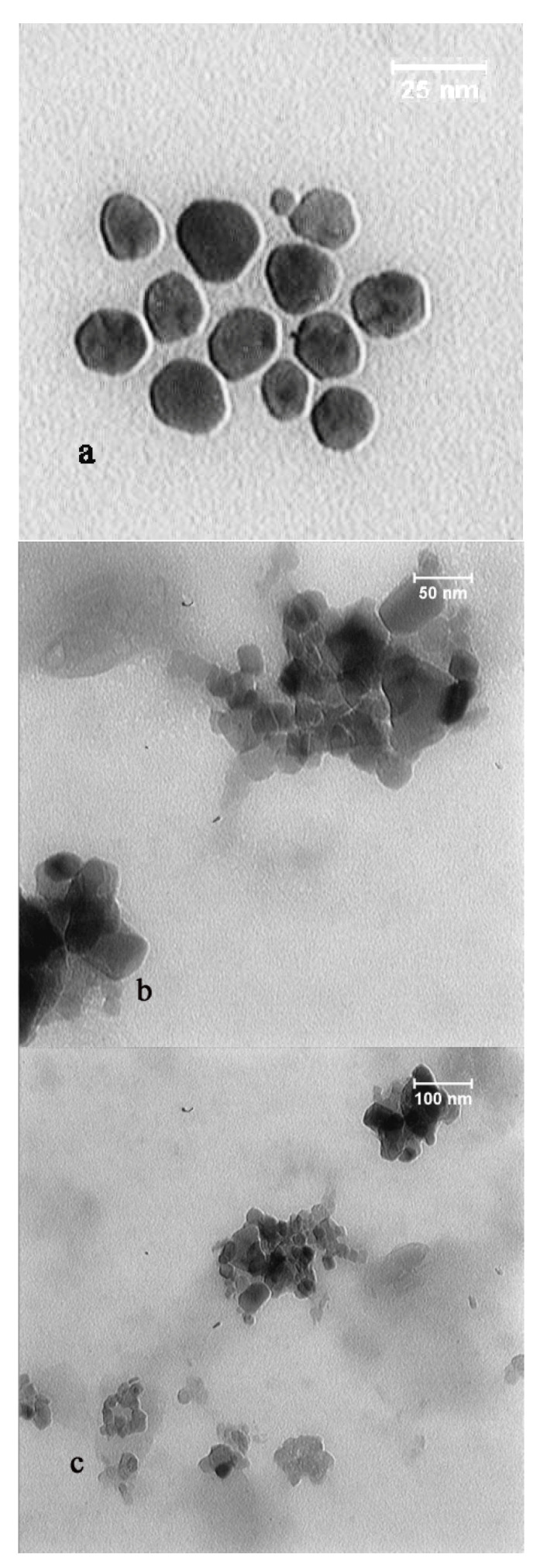
TEM images recorded of an aqueous solution incubated with *Phoma* sp. mycelial biomass with Au+ ions for 48 h in three different scales of (**a**) 25, (**b**) 50 and (**c**) 100 nm. TEM images showed spherical Au particle structures.

**Figure 7 molecules-27-01181-f007:**
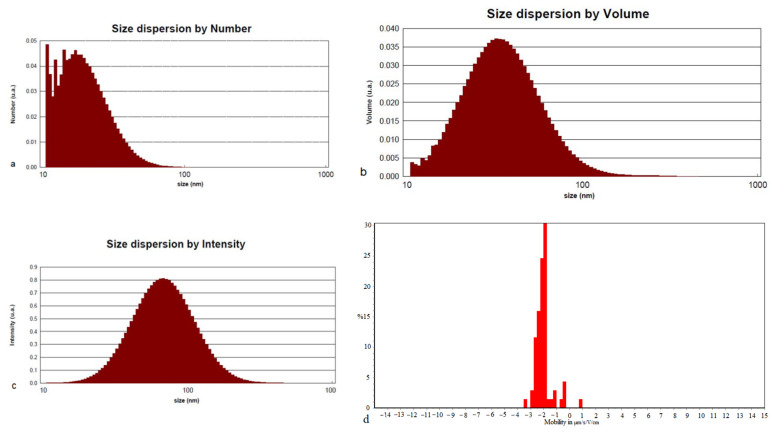
Dynamic light scattering: (**a**) is micrograph that shows size dispersion according to number, Dmean is 20.73, (**b**) is size dispersion by volume, Dmean is 39.65, (**c**) size dispersion by intensity, Dmean is 76.18 and (**d**) show zeta potential measurement of the AuNPs suspension.

**Figure 8 molecules-27-01181-f008:**
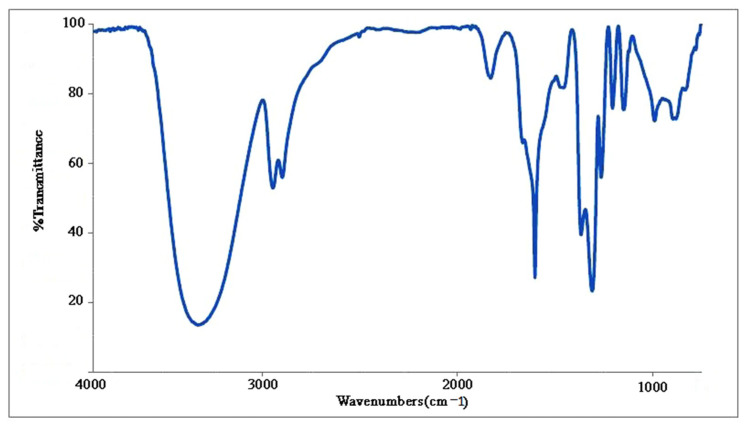
FTIR spectrum of AuNPs synthesized using *Phoma* sp. showing peaks at various regions.

**Figure 9 molecules-27-01181-f009:**
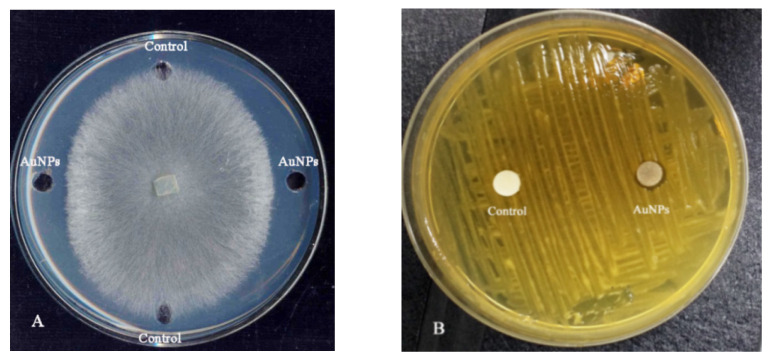
Antifungal and antibacterial activity of biosynthesized AuNPs against plant pathogenies (**A**) well-diffusion methods for *Rhizoctonia solani* AG1-IA, and (**B**) *Xanthomonas oryzae* pv. *oryzae* by disk diffusion method.

**Figure 10 molecules-27-01181-f010:**
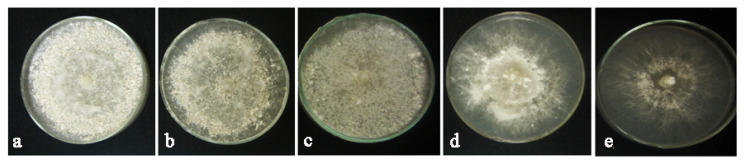
Inhibitory effects of different concentrations of AuNPs on sclerotia formation against *Rhizoctonia solani* AG1. (**a**), (**b**), (**c**) and (**d**) are 10, 20, 40 and 80 concentrations, respectively, and (**e**) is the control.

**Figure 11 molecules-27-01181-f011:**
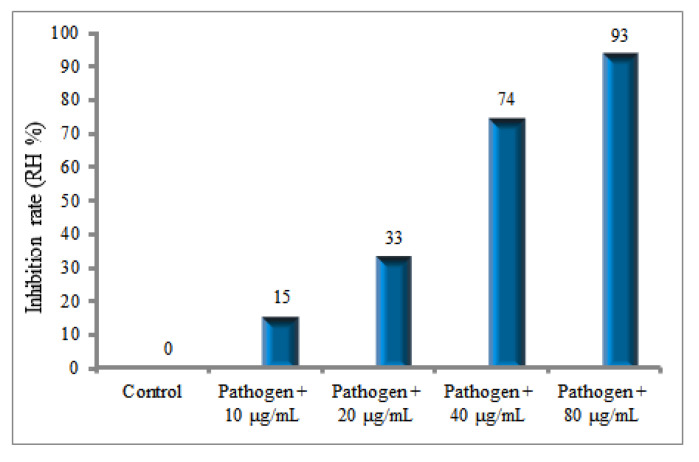
In vitro RH of AuNPs effects on sclerotia formation of *R. solani* AG1.

## Data Availability

The data presented in this study are available in article.

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
