# Peer review of "Evaluation of Phoma sp. Biomass as an Endophytic Fungus for Synthesis of Extracellular Gold Nanoparticles with Antibacterial and Antifungal Properties"

_molecules, 2022, doi:10.3390/molecules27041181_

Round 1

Reviewer 1 Report

1. Some important result datas should be added in the abstract section.

2. The structure and presentation of introduction should be improved.

3. The data analysis method should be added in the part of materials and methods.

4. The results should be presented and described in detaily in the results section. The introduction of results was simply.

5. The effect mechanism should be  improved by cited more related articles.

6. The English language in the manuscipt should be revised and improved.

Author Response

Responses to the comments

We have revised the manuscript according to reviewer’s comments. Listed below are responses to the comments. All the changes addressed in the text as well.

To Reviewer 1

  1. Yes, it’s correct but according to the journal format the abstract should be a total of about 200 words maximum so we can’t add more data to it.
  2. Authors did some related changes.
  3. Recorded data were subjected to analysis with SAS software (SAS Institute, version 9).
  4. Details of DLS and ZETA added.
  5. The effect mechanism of AuNPs antimicrobial added as reviewer suggested.
  6. English quality improved entire the text.

We hope that after these enhancements the manuscript can now be accepted for publication, although we are certainly willing to consider further changes if necessary.

Reviewer 2 Report

The article presented  for review is an excellent study of new application possibilities of gold nanoparticles. The authors have presented in a very professional and in-depth way the process of obtaining raw materials for nanomaterials preparations, obtaining nanoparticles and their characteristics. The paper is fully suitable for publication in such reputed journals. Nevertheless, for the final publication of the research, the authors should clarify several inaccuracies and correct editing errors. A list of noted comments is presented below:

1) Please read the whole article very carefully again - there are unnecessary spaces in the text, lack of superscript in some places, capital letter at the beginning, lack of text alignment.

2) In figure 1e and f there is no scale with the size described in the caption of the figure.

3) In the majority of the figures the axes are illegible!

4) Why the authors presented the particle sizes measured by DLS method in three modes (intensity, size, volume) - if you decided to present it this way please explain what the differences are and what they imply. Observed lack of more extensive commentary on both size and Zeta potential.

5) In line 154 the authors wrote: "Figure 8 shows the FTIR results of pure culture of Phoma sp and the synthesized AuNPs (...)". So why is only one spectrum shown in the figure? Please clarify the description as it does not match the description of sample preparation for FTIR and the results presented.

6) The description of the results of section 2.9 (lines 174-176) is inaccurate, illogical and laconic. Please elaborate.

7) In lines 285-286 the authors briefly describe a possible mechanism for the synthesis of gold nanoparticles. What then happens to the gold chloride? Does it remain in the resulting mixture? Does it not affect the performance of the nanoparticles and their antimicrobial properties? Shouldn't the final product undergo purification processes before application?

Author Response

Responses to the comments

We have revised the manuscript according to reviewer’s comments. Listed below are responses to the comments. All the changes addressed in the text as well.

To Reviewer 2

  1. According to your suggestion we have already edited entire text accordingly.
  2. The scale was added to figure1e-f.
  3. Third comment addressed.
  4. DLS analysis in three modes shown the average particle size of the synthesised displaying high monodispersity. The zeta potential analysis confirmed that the AuNPs biosynthesised are highly stability in water and do not aggregate in solution due to their negative polarity. (This explains was added to section 2.6 of results).
  5. It’s corrected. The FTIR result was just for AuNPs synthesized.
  6. The MBC test was removed and MIC corrected to 160.6 μg/mL. The bacterium was cultured in LB broth medium overnight at 28 °C. About 100 μL of each AuNPs (concentration of 1000 μg/mL) was added to 100 μL Mueller Hinton broth medium to obtain 500 μg/mL of the AuNPs in the first well of each row in 96-well micro-plates. Then AuNPs were serially 2X diluted in 96- well micro-plates containing 1% of DMSO to obtain a range of concentrations from 500 to 0.48 μg/mL in a total volume of 200 μL. Subsequently, teen microliter of bacterial culture (~ 10 7 CFU/mL) was inoculated into each well. The micro-plate was incubated for a day at 28 °C. The MIC was measured as the lowest concentration of the AuNPs at which no measurable growth occurred. All the experiments were done in triplicate.
  7. The little information is available about mechanism for synthesis of gold nanoparticles and our results are primary steps. The gold chlorides remain in the mixture but its rate is low. If in mixture stabilizer and or capping agent were not available, Au0 starts to precipitate. Our antimicrobial tests are based on previously research. According previously research, the antibacterial of AuNPs was demonstrated. Smaller nanoparticles release Au+ faster due to their greater specific surface area. When bacteria are exposed to the nanoparticles, the Au+ released is evenly distributed around the bacteria. They penetrate cell walls and enter the cells. Disulfide bridging between cysteine residues is involved in active oxygen cane be oxygen-containing free radicals or forms of peroxide without free radicals. It is formed in basic metabolism. Reactive oxygen maintained at an appropriate level has a positive effect on cells.

We hope that after these enhancements the manuscript can now be accepted for publication, although we are certainly willing to consider further changes if necessary.

Round 2

Reviewer 1 Report

The presentation of the result including the important data should be improved.

Reviewer 2 Report

The authors addressed most of the comments as expected, so I believe the article can be published as presented.